# Secreted Metabolome of ALS-Related hSOD1(G93A) Primary Cultures of Myocytes and Implications for Myogenesis

**DOI:** 10.3390/cells12232751

**Published:** 2023-11-30

**Authors:** Roberto Stella, Raphael Severino Bonadio, Stefano Cagnin, Roberta Andreotti, Maria Lina Massimino, Alessandro Bertoli, Caterina Peggion

**Affiliations:** 1Istituto Zooprofilattico Sperimentale delle Venezie, 35020 Legnaro, Italy; 2Department of Biology, University of Padova, 35131 Padova, Italystefano.cagnin@unipd.it (S.C.); 3CIR-Myo Myology Center, University of Padova, 35131 Padova, Italy; 4Department of Biomedical Sciences, University of Padova, 35131 Padova, Italyalessandro.bertoli@unipd.it (A.B.); 5Neuroscience Institute, Consiglio Nazionale delle Ricerche, 35131 Padova, Italy; marialina.massimino@cnr.it; 6Padova Neuroscience Center, University of Padova, 35131 Padova, Italy

**Keywords:** amyotrophic lateral sclerosis, SOD1, myogenesis, myocytes, metabolomic profiling, neuromuscular disorders

## Abstract

Amyotrophic lateral sclerosis (ALS) is a motor neuron (MN) disease associated with progressive muscle atrophy, paralysis, and eventually death. Growing evidence demonstrates that the pathological process leading to ALS is the result of multiple altered mechanisms occurring not only in MNs but also in other cell types inside and outside the central nervous system. In this context, the involvement of skeletal muscle has been the subject of a few studies on patients and ALS animal models. In this work, by using primary myocytes derived from the ALS transgenic hSOD1(G93A) mouse model, we observed that the myogenic capability of such cells was defective compared to cells derived from control mice expressing the nonpathogenic hSOD1(WT) isoform. The correct in vitro myogenesis of hSOD1(G93A) primary skeletal muscle cells was rescued by the addition of a conditioned medium from healthy hSOD1(WT) myocytes, suggesting the existence of an in trans activity of secreted factors. To define a dataset of molecules participating in such safeguard action, we conducted comparative metabolomic profiling of a culture medium collected from hSOD1(G93A) and hSOD1(WT) primary myocytes and report here an altered secretion of amino acids and lipid-based signaling molecules. These findings support the urgency of better understanding the role of the skeletal muscle secretome in the regulation of the myogenic program and mechanisms of ALS pathogenesis and progression.

## 1. Introduction

Amyotrophic lateral sclerosis (ALS) is a multifactorial disorder characterized by progressive motor neuron (MN) degeneration, muscle weakness, and wasting. While the vast majority of ALS cases are described as sporadic (sALS), 10% of ALS patients have a family history (fALS) linked to mutations in increasingly recognized genes, the most important of which being C9ORF72 [1,2]; superoxide dismutase 1 (SOD1) [3,4,5,6,7,8]; and TARDBP and FUS [9,10,11,12,13,14]. Albeit less prevalent than sALS, fALS historically has played a significant role in the understanding of ALS pathologic mechanisms, also thanks to studies on mutations of the human (h) SOD1 gene (among which the most-studied is the first identified ALS-related G93A mutation), mainly involving the usage of transgenic (Tg) mice.

The neuromuscular junction (NMJ) is a unique tripartite chemical synapse in which terminal Schwann cells (TSCs), postsynaptic skeletal muscle, and presynaptic MN communicate in unison [15,16]. One of the early events in ALS is related to NMJ disruption, leading to the loss of effective neuromuscular transmission [17], and while the underlying mechanisms remain elusive, growing evidence suggests the synergistic effect of different routes in NMJ dismantling. The first one envisages that MN is the only main character in neuromuscular denervation, while in the second scenario, the skeletal muscle is a fundamental coactor of synaptic degradation, retrogradely participating in MN dysfunction and death [18,19,20,21,22,23]. However, given the critical role TSCs play in maintaining the NMJ and in stimulating nerve terminal regeneration [15,24], the fact that the impaired TSC function contributes to ALS progression is suggested by different data [17,25]. Several findings support the idea that skeletal muscle plays a major role in ALS, including those showing that dysregulation of skeletal muscle metabolism and atrophy occur before MN degeneration in ALS animal models [26,27,28,29,30,31]. Supporting the involvement of the second route, it has also been reported that muscle precursor cells derived from skeletal muscle biopsies of ALS patients show both altered morphology and impaired regenerative capabilities [32,33], and, even more remarkably, the muscle-specific overexpression of mutant hSOD1 is sufficient to produce a reliable panel of ALS disease phenotypes in a Tg mouse model [34,35,36,37,38]. Accordingly, myogenic precursor satellite cells from hSOD1(G93A) Tg mice and the hSOD1(G93A)-overexpressing C2C12 myoblast cell line have an impaired proliferation and an alteration of myogenic process, respectively [39,40]. Despite the aforementioned findings, the role of skeletal muscle in ALS as a mere victim of MN disease or a conspiratorial enemy is still debated.

It is widely recognized that skeletal muscle is capable of modulating several processes and communicating with other tissues and organs, regulating their metabolism and playing a role in inflammatory processes, angiogenesis, and also in self-regulated myogenesis [41,42,43,44,45], through the production and secretion of several mediators enabling their communication in an auto-, para-, or endocrine way [45,46]. Although the best-known example of secreted mediators is the family of myokines, several other types of molecules are released by skeletal muscle cells, such as lipids, amino acids, metabolites, and small RNAs [47]. While it is widely recognized that skeletal muscle secretome changes as an effect of exercise, aging, or different pathophysiological conditions, such as sarcopenia or diabetes [41,42,47,48,49], thus influencing lipid and glucose metabolism [49,50], little is known about skeletal muscle secretome changes in ALS [51,52]. This prompts the need for understanding if and how muscle cell-secreted factors play a role in homo- and heterocellular communication during ALS genesis and progression.

Here, we show that the in vitro myogenesis of hSOD1(G93A) primary skeletal myocytes, in which myoblasts were left proliferating (48 h) before inducing differentiation into self-contracting myotubes by serum withdrawal [53], is impaired in comparison to hSOD1(WT) controls, supporting previous findings based on different paradigms [39,40,54].

Encouraged by such a phenomenological observation, we chose to analyze the ALS skeletal muscle-secreted metabolome, using isolated primary cultured skeletal myocytes to exclude any effect due to the presence of other cell types. We performed a comparative characterization of hSOD1(G93A) and hSOD1(WT) myocyte cultures by using liquid chromatography coupled with high-resolution mass spectrometry (LC–HRMS) at two different time points of in vitro culturing (i.e., at the end of the proliferation step and after differentiation). This approach allowed us to unveil a profound alteration in the abundance of a group of secreted molecules at the first time point (proliferative), including metabolites related to the pathways of amino acids and glycerolipid and pyrimidine metabolism.

Importantly, we also showed that the altered myogenesis of hSOD1(G93A) myocytes was rescued by the addition of the conditioned proliferation medium from the corresponding healthy hSOD1(WT) control cells. Such an observation substantiated that secreted molecules from healthy myocytes were able to correct the defective myocyte differentiation in the pathologic setup, providing a little step toward new avenues for the identification of therapeutic molecules for preventing or limiting muscle alterations in ALS patients.

## 2. Materials and Methods

### 2.1. Mouse Models

hSOD1(WT) (strain B6SJL(Tg-SOD1)2Gur/J) and hSOD1(G93A) (B6SJL(Tg-SOD1*G93A)1Gur/J mice) Tg mice were from Jackson Laboratories (cat. n. 002297 and 002726, respectively). The mice strains overexpress the hSOD1 protein or the wild-type (WT) or carry the ALS-related G93A point mutation, respectively. Colonies were maintained by breeding Tg hemizygote males with wild-type B6SJLF1/J hybrid females. Newborns were genotyped as previously described [55] using the MyTaq Extract-PCR kit (Bioline).

### 2.2. Primary Myocyte Cultures and Myocyte-Conditioned Medium (MCM) Preparation

Primary cultures of skeletal myocytes were prepared as previously described [56] from P1-3 mice. Briefly, after dissection of the posterior limb muscles, tissues were minced and enzymatically disaggregated by incubating the tissue in a buffer containing trypsin 0.1% (*w*/*v*) diluted in phosphate-buffered saline (PBS; 140 mM NaCl, 2 mM KCl, 1.5 mM KH_2_PO_4_, 8 mM Na_2_HPO_4_, and pH 7.4) for a total time of 45 min at 37 °C. After filtration and centrifugation, cells were resuspended in a proliferation medium (Ham’s F12 with 10% fetal bovine serum, 2 mM glutamine, 100 U/mL penicillin, and 100 µg/mL streptomycin) for 2 days (2 Days In Vitro: 2 DIV). At this time point, the proliferation medium was removed and then replaced by differentiation medium (Dulbecco’s Modified Eagle Medium containing 2% horse serum, 2 mM glutamine, 100 U/mL penicillin, and 100 µg/mL streptomycin) for myocyte differentiation into myotubes. Cells were seeded at a density of 2.5 × 10^5^ cells onto 13 mm coverslips coated with collagen (0.1% *w*/*v* in PBS) for immunocytochemistry or at a density of 1 × 10^6^ cells onto 35 mm collagen-coated plates for biochemical assays. All reagents for cell culture were from Euroclone (Milan, Italy).

Myocyte-conditioned medium (MCM) was prepared by collecting culture medium from hSOD1(WT) and hSOD1(G93A) primary myocytes after 2 DIV (proliferation stage), and after 4 DIV (differentiation stage). MCM was immediately stored at −80 °C after being centrifuged at 16,000× *g* (4 °C, 5 min) to remove cell debris.

Additionally, hSOD1(G93A) primary myocytes were cultured in the presence of either MCM collected at 2 DIV from hSOD1(WT) or from hSOD1(G93A) cell cultures, which was added immediately after seeding at a final concentration (f.c.) of 20% (*v*/*v*) and maintained only for the proliferation stage (2 DIV). As described above, the proliferation medium was replaced by a differentiation medium after 2 DIV, and cells were analyzed at 4 DIV by using immunocytochemistry or Western blot (WB).

### 2.3. Phenotypic Characterization of Myotubes

The surface area of myocytes at 4 DIV was assessed by measuring the surface area of myotubes by taking images with a phase-contrast microscope (Axiovert 100, Zeiss) equipped with a computer-assisted charge-coupled camera (AxioCam, Zeiss) at 10 × magnification. Images were analyzed using ImageJ software [57]. Differentiated myotubes were recognized by their elongated tubular shape. The myotube area was measured by using ImageJ software (Version 1.53t of 24 August 2022). For determining the area frequency distribution, myotubes with a surface area between 0 and 100,000 pixels were split into twenty intervals of 5000 pixels, and the frequency of myotubes in each interval was calculated.

### 2.4. Immunocytochemistry

4 DIV primary cultures grown on coverslips were rinsed twice with phosphate-buffered saline (PBS) and fixed with paraformaldehyde (2% (*w*/*v*) in PBS, 20 min, 4 °C). Cells were permeabilized in Triton-X100 (0.5% (*w*/*v*) in PBS (15 min, RT) and then incubated (overnight, 4 °C) with mouse monoclonal antibody anti-desmin (mAb) (Boehringer, cat. no. 814377; 1:50 dilution in PBS containing 1% (*w*/*v*) bovine serum albumin (BSA), PBS–BSA). After extensive washings in PBS, cells were incubated (1 h, RT) with Alexa Fluor 555-conjugated anti-mouse IgG (Molecular Probes, Thermo Fisher Scientific, Waltham, MA, USA, cat. no. A21424, 1:500 dilution in PBS–BSA) and finally counterstained with the Hoechst 33342 nuclear fluorogenic probe (Sigma-Aldrich, St. Louis, MO, USA, 5 µg/mL in PBS, 20 min, RT). After further washings in PBS, the coverslips were mounted onto microscope slides using a fluorescence mounting medium (DAKO) and observed with an inverted fluorescence microscope (Axiovert 100, Zeiss) equipped with a computer-assisted charge-coupled camera (AxioCam, Zeiss) at 10 × magnification. Images from different fields were digitalized and stored for subsequent analysis. 

The differentiation index was calculated as the ratio between the sum of myonuclei and the total number of desmin (a myocyte marker)-positive cells.

The fusion index was calculated as the ratio between the number of total nuclei inside desmin-positive myotubes with more than three nuclei and the total number of myonuclei. 

Counts for both the fusion index and the differentiation index were performed in 16 randomly selected fields from four different biological replicates and reported as mean ± SEM of four different cultures for each condition.

### 2.5. Western Blot Analysis

After two washes with ice-cold PBS, primary cultures of myocytes at either 2 DIV or 4 DIV were lysed in an ice-cold lysis buffer containing 10% (*w*/*v*) glycerol, 2% (*w*/*v*) SDS, 62.5 mM Tris/HCl, pH 6.8, and protease and phosphatase inhibitor cocktails then centrifuged (14,000× *g*, 10 min, 4 °C) to pellet cell debris. The total protein content in the supernatant was determined by using the bicinchoninic acid assay kit (Thermo Fisher Scientific, Waltham, MA, USA) according to the manufacturer’s instructions. Protein samples were then diluted to the desired concentration by adding dithiothreitol (f.c., 50 mM) and bromophenol blue (f.c., 0.004% (*w*/*v*)) and boiled (5 min). Proteins were separated by using SDS-PAGE using Mini-Protean TGX precast gels (4–20%, Bio-Rad Laboratories, Hercules, CA, USA) and then transferred onto polyvinylidene difluoride (PVDF) membranes (0.45 µm pore size; Bio-Rad Laboratories). Membranes were incubated in Tris-buffered saline (TBS, 20 mM Tris-HCl, pH 7.6, and 150 mM NaCl) added with 0.1% (*w*/*v*) Tween-20 (TBS-T) and 3% (*w*/*v*) BSA (Sigma-Aldrich) as blocking solution (1 h, RT) and then incubated with the desired primary Ab diluted in blocking solution (overnight, 4 °C). After washing in TBS-T, membranes were incubated (1 h, RT) with horseradish peroxidase-conjugated anti-mouse or anti-rabbit (Sigma-Aldrich, cat. no. A9044 and A0545, respectively) depending on the primary Ab used. After washings with TBS-T, immunoreactive bands were visualized and digitalized using a digital camera workstation (NineAlliance, UVITEC, Eppendorf, Hamburg, Germany), using an enhanced chemiluminescence reagent kit (Millipore). For densitometric analyses, the optical density of each immunoreactive band was normalized to the optical density of the corresponding Coomassie blue (50% (*v*/*v*) methanol, 7% (*v*/*v*) acid acetic, and 0.01% (*w*/*v*) Coomassie blue)-stained lane [58]. For the analysis of phosphorylated proteins, samples were run in parallel onto the same gel, and PVDF membranes were probed with antibodies recognizing either the phosphorylated or the total form of the protein of interest. Densitometric values of each phosphorylated protein were normalized to those of the corresponding total protein band (calculated as above).

The following primary Abs were used: Anti-EmbMyHC mouse mAb (cl.BF-G6), 1:5000 (Developmental Studies Hybridoma Bank, University of Iowa); Anti-p38 rabbit polyclonal antibody (pAb), 1: 1000 (Cell Signaling Technology Danvers, MA, USA; cat. no. 9212); Anti-pThr180/Tyr182 p38 rabbit mAb, 1:1000 (Cell Signaling Technology; cat. no. 9211); and Anti-MyoD mouse mAb (cl. 5.8A), 1:1000 (Thermo Fisher Scientific, cat. no. MA5-12902).

### 2.6. Metabolomic Analysis of the Secretome of Primary Cultures of Myocytes

Metabolomics of collected MCM derived from hSOD1(WT) and hSOD1(G93A) primary myocytes was performed as in [59]. Briefly, 2 DIV and 4 DIV MCM were collected and centrifuged (10 min, 16,000× *g*, 4 °C), then proteins were precipitated by adding 4 volumes of methanol to 1 volume of MCM and incubating the solution for 16 h at −80 °C after mixing. After centrifugation (15 min, 16,000× *g*, 4 °C), the pellet was discarded and an aliquot of 150 µL of each supernatant was transferred into a new tube and dried at room temperature under a stream of nitrogen. Dried MCM were then resuspended with 75 µL of acetonitrile/water 50/50 (*v*/*v*) and analyzed by using liquid chromatography and high-resolution mass spectrometry (LC–HRMS). The instrumental setup consisted of an ultra-high performance liquid chromatography system (UHPLC, Ultimate 3000, Thermo Scientific) coupled with a hybrid quadrupole-orbitrap mass spectrometer (Q-Exactive, Thermo Scientific). Separation of metabolites was obtained by injecting samples onto a Zic-HILIC column (100 × 2.1 mm; 3.5 µm particle size, Merck, Darmstadt, Germany) maintained at 30 °C. Mobile phases consisted of 2.5 mM acetic acid and 2.5 mM ammonium acetate in water, pH 6.0 (solvent A) and acetonitrile (solvent B) at a flow rate of 0.3 mL/min. The gradient elution profile combining solvent A and solvent B was as follows (%A, t (min)): (5%, 0–2 min), (5–60%, 2–15 min), (60%, 15–19 min), (60–5%, 19–19.5 min), and (5%, 19.5–25 min).

Metabolomics profiling of MCM samples was conducted both in positive and negative ionization polarity. Source parameters were as follows: spray voltage, 2.5 kV and −2.5 kV, respectively, for positive and negative ionization polarity; capillary temperature, 325 °C; sheath gas flow rate, 40 arbitrary units (a.us.); auxiliary gas flow rate, 10 a.us.; S-lens voltage, 50 V; in-source fragmentation voltage, 0; and heater temperature, 325 °C. Data were acquired in full-scan acquisition mode from 70 to 1000 *m*/*z* with a resolving power of 70,000 full width at half maximum at 200 *m*/*z*.

To exclude potential instrumental drift, we repeatedly injected a quality control (QC) sample (prepared by mixing an equal volume of supernatant from each sample under investigation and by adding labelled leucine-5,5,5-D3 and L-tryptophan-2,3,3-D3 at a final concentration of 1 ng/μL) throughout the analytical session.

We carried out relative quantification and comparison of the metabolic fingerprint by using Compound Discoverer software (Thermo Fisher Scientific, version 2.1) to align chromatographic peaks, integrate compound peaks, and normalize area values. 

### 2.7. Metabolomics Data Analysis

Data obtained from negative or positive ionization polarity were initially checked by performing an unsupervised multivariate principal component analysis (PCA). Altered metabolites were determined as in [59], by inspecting and filtering chromatographic peaks and by excluding metabolites displaying a relative standard deviation above 30% in QC samples. 

The relative amount of each metabolite was expressed as normalized chromatographic peak area value. We calculated the ratio between the amount measured in hSOD1(G93A) and that of hSOD1(WT) samples. A Wilcoxon nonparametric test was performed for statistics in the comparison of hSOD1(G93A) and hSOD1(WT) MCM. Volcano plots were built using Excel software (Microsoft 365) to show the magnitude (fold change) and statistical significance of altered metabolites. For each compound, the fold change expressed as log_2_ (G93A/WT) on the x-axis and −log_10_ (*p*-value) on the y-axis was plotted. Only metabolites showing a fold change ≥ |2| (corresponding to a G93A vs WT ratio <0.25 and >4) and a −log_10_ (*p*-value) > 3 (corresponding to a *p*-value < 0.001) were considered as significantly altered.

Pathway analysis was performed using the free web-based software, MetaboAnalyst (Version 5.0) [60].

### 2.8. Statistical Analysis

Western blot and immunohistochemistry data were analyzed using GraphPad (version 8.0.2) software. Statistical analyses were performed as indicated in the figure legends by using the Mann–Whitney nonparametric test or one-way ANOVA analysis when comparing three groups after the determination of the normal distribution of data. n indicated the number of biological replicates, considered independent primary cultures. The significant level was established at *p* < 0.05. Metabolomics data were analyzed by using PCA using SIMCA-P software (version 13.0) and Wilcoxon nonparametric test using ORIGIN software (version 2018) considering significant *p* < 0.001.

## 3. Results

### 3.1. Impaired Myogenesis in hSOD1(G93A) Myocytes Compared to That of hSOD1(WT) Primary Cells

Starting from previous findings showing ALS-related alterations in myogenesis [39,40,54], we decided to evaluate morphometric parameters and the expression of the differentiation marker myosin heavy chain-embryonic (EmbMyHC) in primary cultures derived from hSOD1(G93A) mice and the hSOD1(WT) counterpart. Bright-field images evidenced that at 4 DIV of culturing, hSOD1(G93A) primary myotubes were smaller and thinner relative to controls, as evidenced by the absence of large myotubes (indicated with asterisks in the upper panel of Figure 1A) in the hSOD1(G93A) primary culture.

This preliminary morphological observation was supported by measuring the size frequency distribution and surface area of myotubes (Figure 1B,C, respectively). Interestingly, biomolecular approaches showed a significantly reduced expression of EmbMyHC at 4 DIV in hSOD1(G93A) primary myocytes, further suggesting an impaired ability of ALS-related myocytes to differentiate (Figure 1D).

### 3.2. MCM Analysis by Using Liquid Chromatography–High Resolution Mass Spectrometry (LC–HRMS)

Skeletal muscle is known to influence the myogenic program by itself thanks to the secretion of several metabolites. In previous work, we set up a procedure allowing us to unveil significant alterations in the secreted metabolome of primary cultures of spinal astrocytes derived from hSOD1(WT) and hSOD1(G93A) mice [59]. We hypothesized that different soluble factor secretion could account for the altered myogenic phenotype described in Figure 1. For this reason, we applied the same analytical approach to the culturing medium (indicated also as MCM) of hSOD1(WT) and hSOD1(G93A) muscle precursor cells collected at the end of the proliferation step (2 DIV) and during the differentiation step (4 DIV).

In summary, we detected and relatively quantified 1297 and 1141 compounds in the positive and negative ionization modes at 2 DIV, respectively, while 1599 and 940 compounds were detected and relatively quantified in the positive and negative ionization modes at 4 DIV, respectively. Interestingly we found that at 2 DIV, the secreted metabolomes of the two primary culture populations with different hSOD1 genotypes were profoundly different. Indeed, as shown in Figure 2A,B, the PCA score plots based on data acquired in the positive or negative ionization mode displayed two independent and clustered groups. These two distinct groups correspond to conditioned medium derived from hSOD1(WT) (blue circle) or hSOD1(G93A) (green hexagon) myocytes, thus revealing a distinct globally secreted metabolome according to the hSOD1 genotype. Remarkably, at 4 DIV, no difference was detected between the two populations of primary cultures of skeletal myocytes, as depicted in the PCA score plots of Figure 2C,D.

### 3.3. MCM from hSOD1(WT) Myocytes Rescued the Myogenic Differentiation of hSOD1(G93A) Cells

Having found such profound differences in the 2 DIV MCM between the hSOD1 genotypes, we asked whether the hSOD1(WT) 2 DIV secreted factors influenced the myogenic program of hSOD1(G93A) cells. To this purpose, we cultured hSOD1(G93A) primary myocytes in the presence of 2 DIV MCM from hSOD1(WT) myocytes (20%, *v*/*v*) from the seeding of precursor cells until the end of the proliferation stage (2 DIV) and then analyzed different biochemical and morphometric indexes of in vitro myogenesis at 4 DIV.

Our data indicate that such an addition was sufficient to make hSOD1(G93A) myocytes recover their intrinsically altered myogenesis, as demonstrated by EmbMyHC expression (Figure 3A) and the differentiation and the fusion indexes (Figure 3B,C, respectively).

The myoblast determination protein (MyoD) is a well-known myogenic transcription factor that regulates the differentiation potential of activated myoblasts [61,62]. To investigate early mechanisms in the rescue of hSOD1(G93A) in vitro myogenesis, we compared the expression of MyoD in such cells at 2 DIV in the absence or the presence of hSOD1(WT), added immediately after cell seeding. The increased abundance of MyoD strongly suggests that healthy myocytes secrete molecules able to restore the differentiation program of ALS-related myocytes during the first 2 days of culturing (Figure 4A). Accordingly, under the same experimental setting, p38 mitogen-activated protein kinase (MAPK), which plays a central role in myogenesis [63], was shown to be significantly more activated in hSOD1(G93A) by the addition of WT MCM (Figure 4B).

### 3.4. Analysis and Annotation of Altered Metabolites in hSOD1(G93A) MCM

We next performed a thorough analysis of LC–HRMS data in an attempt to understand how the MCM from healthy myocytes might be able to correct the myogenic defect in ALS-related cells.

Among the 1297 and 1141 compounds detected and relatively quantified in the positive and negative ionization modes, the abundance of 11 (detected in the positive ionization mode) and 15 molecules (detected in the negative ionization mode) were found to be significantly altered in the hSOD1(G93A)-conditioned medium compared to the hSOD1(WT) counterpart. Such an overall result is depicted in the two volcano plots of Figure 5A,B in which the statistical significance vs the fold change is reported for all detected compounds. Molecules found to be significantly altered were annotated by using the CEU Mass Mediator tool [64] based on the measured *m*/*z* value of the precursor ion and using MassBank [65]. Most of the significantly altered metabolites were annotated (i.e., 23 out of 26), and about half of which were short peptides of three or four amino acids (Table 1).

Two alternative analyses, i.e., Metabolite Set Enrichment Analysis (Figure 6A) and Metabolic Pathway Analysis (Figure 6B), were then performed using the Small Molecule Pathway Database and the KEGG database, respectively [66], to possibly rationalize metabolic differences among MCM subtypes. In both analyses, the most frequently modified pathways were related to amino acid metabolism, including—among others and with particular relevance—methionine, arginine, glutamine, and glutamate. Enrichment and pathway analyses also showed modifications in glycerolipid and pyrimidine metabolism.

## 4. Discussion

In this paper, we used primary cultures of myoblasts derived from muscle precursor cells of newborn mice to study in vitro myogenesis, allowing us to confirm previous findings of impaired myogenesis in different ALS models, such as C2C12 cells overexpressing SOD1(G93A) or mutant SOD1 iPSC-derived myotubes [31,39,54,67].

Myogenesis is a highly complex, finely regulated process in which myoblasts (derived from mesodermal cells during embryonic development or from satellite cells during adult skeletal muscle regeneration) proliferate, exit the cell cycle to enter a full differentiation program and form polynucleated syncytia up to mature skeletal muscle fibers [68]. Skeletal muscle atrophy and impaired function are evident consequences of MN death and NMJ breakdown typically observed in ALS, to which TSC-impaired functionalities have also been demonstrated as contributors [17,25]. Nonetheless, it has also been shown that skeletal muscle damage may in turn promote MN demise, particularly in SOD1-related ALS forms, as a result of dying-back mechanisms [69].

We set up primary cultures of myocytes from hSOD1(G93A) and hSOD1(WT) control newborn mice, representing a presymptomatic skeletal muscle model of the disease, to explore the possibility that altered myogenesis is an intrinsic defect in ALS early stages rather than a simple consequence of MN loss. By using this model, we first showed a delayed in vitro differentiation of hSOD1(G93A) primary myoblasts, as demonstrated by both the alteration of morphometric parameters (i.e., dimension of myotubes) and the expression of EmbMyHC, a marker of skeletal muscle differentiation.

Starting from the notion that secreted soluble factors play fundamental roles in biology, regulating an unpredictable number of pathophysiological processes, including myogenesis itself, we compared the medium collected from hSOD1(G93A) and hSOD1(WT) primary myocyte cultures during the proliferation or the differentiation stage using global metabolomic profiling by using LC–HRMS analysis.

The choice of such a metabolomic approach also stemmed from acknowledging that aberrantly secreted soluble factors or extracellular molecules may be markers of intrinsic skeletal muscle dysmetabolism in hSOD1(G93A) myoblasts. In addition, secreted cytokines and peptide hormones have been already associated with skeletal muscle in ALS [36,52,70,71,72,73], further supporting our experimental strategy to scrutinize the extracellular metabolome.

While we did not find any significant difference in MCM during the differentiation phase (i.e., at 4 DIV), and profound alterations were observed in MCM collected during the proliferative phase (i.e., after 2 DIV). Such findings suggest that factors secreted during the first stages of cell proliferation and commitment to differentiation might be those related to the defective myogenic program of hSOD1(G93A) myocytes. Consistent with this hypothesis was the finding that the very early addition (i.e., immediately after isolation and plating) of MCM from 2 DIV healthy myocytes rescued hSOD1(G93A) myoblasts from delayed in vitro myogenesis, as also supported by the increased expression of MyoD and the activation of p38. Both the MyoD and p38 pathways are indeed known to play a critical role in the regulation of myogenesis and muscle cell differentiation [61,62,63]. These results suggest that ALS myocytes suffer from a defective priming toward myogenesis that can be corrected in the first 2 DIV (i.e., 48 h of culturing) by soluble factors secreted by healthy cells.

The comparative analysis of LC–HRMS data identified 26 metabolites differently secreted from 2 DIV hSOD1(G93A) cultured myocytes compared to the hSOD1(WT) counterpart, 23 of which were annotated. In this analysis, we adopted a high cut-off threshold and a stringent p-value to restrict the spectrum of identified metabolites to those that may be quantitatively and statistically relevant in supporting our data on the different phenotypes. In addition, the identification of a restricted, highly statistically confident molecule set would render more feasible a targeted approach whereby selected factors are tested (in vitro and/or in vivo) for their ability to correct skeletal muscle defects in ALS models.

It must be acknowledged, however, that annotated molecules may comprise either factors differentially secreted or culture media elements (including collagen) differentially processed by myocytes with a different hSOD1 genotype. It is also worth emphasizing that differentially present metabolites can be a direct cause, or a consequence, of the defective differentiation phenotype of ALS myocytes, as well as epiphenomena related to other alterations of muscle cell physiology. This point would deserve further studies, as discussed below.

We found strikingly reduced levels of glutamine (which is already present at a 2 mM concentration in the culture medium) in the MCM from ALS myocytes, which also correlates with lower levels of products of glutamine metabolism, such as pyrimidines and purines (see Table 1). Skeletal muscle represents the main tissue for glutamine synthesis, storage, and release [74]. Glutamine has antioxidant (as a precursor of glutathione) and anti-inflammatory properties, modulates the synthesis of heat shock proteins [74], which are implicated in the response to the proteostatic stress in ALS and other neurodegenerative disorders [75], and participates in skeletal muscle differentiation [76]. However, during catabolic situations, as might occur in the muscle of ALS patients [74,77], glutamine concentration in plasma and tissues (especially in skeletal muscle) is severely compromised [78]. Alterations in glycolysis, resulting in reduced levels of acetyl-CoA, have been reported in muscles of a mouse model of ALS [77], which may demand glutamine to contribute (via glutamate) to the overall mitochondrial energetic metabolism by supporting the citric acid cycle through the anaplerotic production of alpha-ketoglutarate. We may therefore reason that higher glutamine uptake from the culture medium and/or reduced glutamate release reflects the attempt of cells to counteract a defective muscle energy metabolism, which may well account for impaired differentiation. Interestingly, one half of the annotated altered metabolites were tri- and tetrapeptides (5–15 folds higher in the culture medium of ALS-related myocytes). Further investigations are needed to uncover whether such results could be explained by a higher degradation of extracellular matrix components. Interestingly, the overexpression of several matrix metalloproteinases was found both in serum [79] and in muscle biopsies of ALS patients [80].

Lysophosphatidic acid (LPA), another molecule we found to be reduced (approximately six fold) in the supernatant of ALS-related myocytes, is a pluripotent lipid mediator involved in extracellular signaling and capable to influence many biochemical processes, both in health and disease [81]. LPA is mainly produced through the activity of autotaxin (ATX), and it has been recently reported that the ATX/LPA/LPA receptor1 axis regulates satellite cell differentiation and muscle regeneration [82]. Moreover, ATX transgenic mice show accelerated muscle regeneration and hypertrophy [82]. The lower levels of secreted LPA in hSOD1(G93A) primary myocytes could thus contribute to their defective differentiative phenotype and the rescue capability exerted by the secretome of healthy cells.

Another molecule we found to be severely downregulated among secreted metabolites by ALS myocytes was prostaglandin F1α (PGF1α), belonging to the prostaglandin (PG) family of cyclooxygenase-dependent arachidonic acid derivatives. In particular, PGs are extremely potent biologic substances produced by nearly all cells of the body [83], including skeletal muscle, which is able to synthesize both E and F subfamily PGs acting as paracrine or autocrine regulators of in vitro [84] and in vivo [85] myogenesis. Over the past years, PGs (in particular, PGE2) have been reported to regulate skeletal muscle adaptations to aging and exercise [86] and skeletal muscle stem-cell function, improving regeneration and strength [87]. Our finding of reduced PGF1α in the hSOD1(G93A) myocyte secretome may thus support the contention that PGs participate in the altered/diminished differentiation/proliferation of ALS myocytes (rescued by WT cells) and possibly alter the regenerative capacity of adult skeletal muscle during disease.

In summary, the stringent constraints of our analysis led to the identification of a few specific metabolites that significantly differ in the MCM from 2 DIV hSOD1(G93A) and hSOD1(WT) primary cultured myocytes. It must be acknowledged, however, that any effective role of such molecules in the auto- or paracrine regulation of the myogenic program is still to be definitively demonstrated. For example, the alteration of the identified metabolites (or at least some of them) could just be an epiphenomenon consequent to an intrinsic inability of ALS myocytes to correctly differentiate. To better clarify this issue, experiments envisaging the specific addition of each of these molecules, or combinations thereof, to primary cultured myocytes would be required. Importantly, if these experiments yielded positive results, the extension of the approach to in vivo ALS models would be quite obvious, possibly enabling the detection of retrograde effects on motor neurons. It is also fair to acknowledge that other secreted molecules or cellular factors could be responsible for the defective hSOD1(G93A) phenotype and its rescue by hSOD1(WT) MCM, including extracellular vesicles, miRNAs, and/or other metabolites that escaped our analysis, a topic that deserves further studies. Finally, it must be considered that in this work, we employed primary myocytes from newborn mice, thus in a very presymptomatic phase. It would be of great interest for future work to ascertain if the same phenotypes (differentiative and secretory) also pertain to adult satellite cells from hSOD1(G93A) mice at symptomatic disease stages because this would reveal an impaired intrinsic regenerative capability of skeletal muscles in ALS.

## 5. Conclusions

This work demonstrates that the hSOD1(G93A)-derived culture medium has a completely different secretome than the hSOD1(WT) counterpart and that the addition of MCM from healthy myocytes was capable of correcting the delayed in vitro myogenic program of ALS-related cells.

We also annotated several soluble metabolites that might act in an autocrine or paracrine manner to regulate myogenesis and potentially play a role in correcting the defective myocyte differentiation in our ALS model. The added value is that the role of such molecules might go far beyond the ALS context, given that they could provide general insights into skeletal muscle biology and novel hints for the mechanisms of other (neuro)muscular diseases.

However, together with the molecules investigated in this study, additional soluble factors of a different nature, such as myokines or miRNAs, may represent new therapeutic molecules aimed at limiting muscle alterations in ALS patients; hence, future research needs to be focused on the characterization of the secreted proteome and released miRNA by hSOD1(WT) myocytes.

## Figures and Tables

**Figure 1 cells-12-02751-f001:**
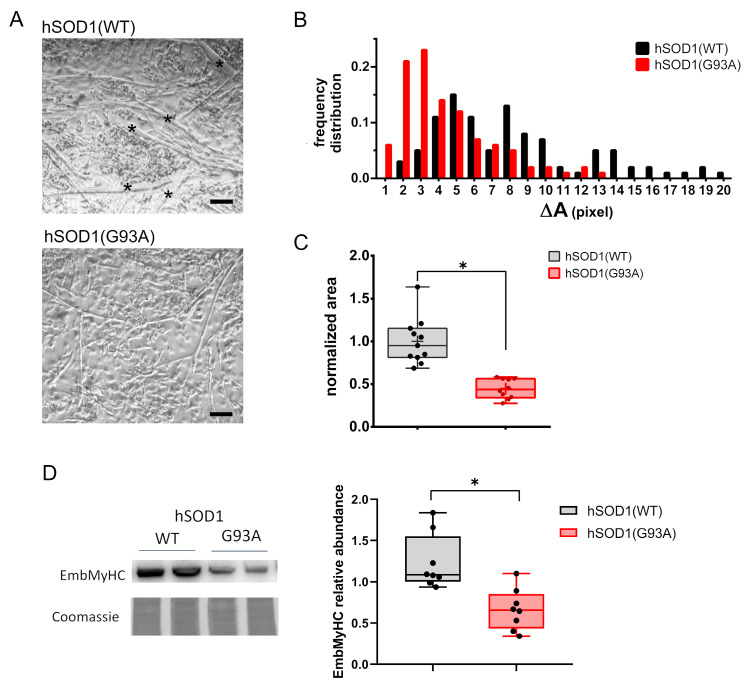
hSOD1(G93A) primary cultures of skeletal myocytes show impaired in vitro differentiation compared to the hSOD1(WT) counterpart. (**A**) The representative phase-contrast images of myotubes obtained from hSOD1(WT) and hSOD1(G93A) after 4 DIV. Myotubes with higher dimensions are indicated by black asterisks. Scale bar = 40 μm. (**B**) The differentiation of hSOD1(WT) and hSOD1(G93A) primary cultured myocytes at 4 DIV was evaluated by the frequency distribution of the myotube area. Myotubes with a surface area from 0 to 100,000 pixels were clustered in groups with a range of 5000 pixels defined as ΔA. A total of 174 hSOD1(WT) and 133 hSOD1(G93A) myotubes were considered, derived from 10 random fields of three different cultures for each genotype of hSOD1. (**C**) The myotube average area was calculated in 11 and 10 different fields of three hSOD1(WT) and hSOD1(G93A) primary cultured myocytes and normalized to the mean value of hSOD1(WT). Here and after, the box plot represents the first and third quartiles, and the inner line and the cross represent the median and the mean, respectively, and the extreme of whiskers represents the outliers. * *p*-value < 0.05, Mann–Whitney *U* test. (**D**) The expression of EmbMyHC was evaluated by using WB in hSOD1(WT) and hSOD1(G93A) primary cultured myocytes at 4 DIV. The left panel shows representative WBs and the corresponding Coomassie blue staining of the membrane, while the graphs on the right report the densitometric analysis. *n* = 8 for each hSOD1 genotype. * *p*-value < 0.05, Mann–Whitney *U* test.

**Figure 2 cells-12-02751-f002:**
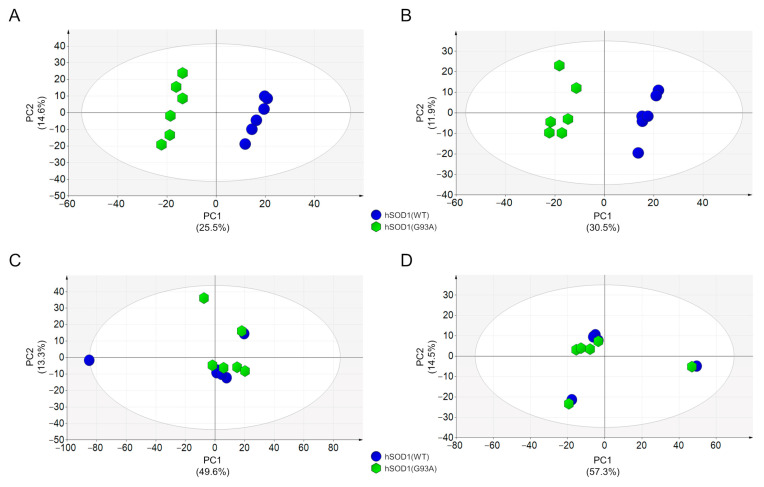
Metabolomic analysis of hSOD1(WT) and hSOD(G93A) MCM. (**A**,**B**) PCA score plots of MCM metabolites from primary myocytes at 2 DIV. Data were acquired by using HILIC-HRMS in positive (**A**) and negative (**B**) ionization modes. (**C**,**D**) PCA score plots of MCM metabolites from primary myocytes at 4 DIV. Data were acquired by using HILIC-HRMS in positive (**C**) and negative (**D**) ionization modes. MCM from hSOD1(WT) myocytes are depicted as blue circles, while MCM from hSOD1(G93A) myocytes are reported as green hexagons. The percentage of variance explained by the first two principal components (PC1 and PC2) is reported in each PCA score plot. Ellipse: Hotelling T2 (95%).

**Figure 3 cells-12-02751-f003:**
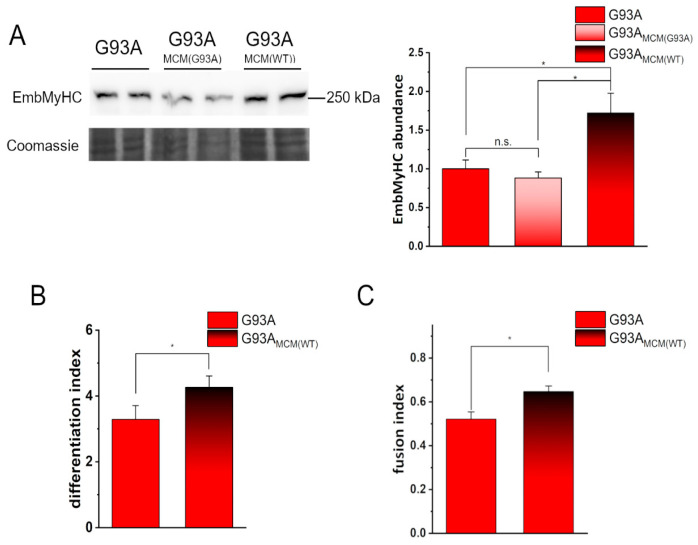
Conditioned medium from hSOD1(WT) myocytes recovers the defective in vitro differentiation of hSOD1(G93A) cells. (**A**) WB analysis of EmbMyHC abundance in hSOD1(G93A) myocytes grown 4 DIV in the absence or the presence of hSOD1(G93A) or hSOD1(WT) MCM (MCM(G93A) and MCM(WT), respectively, as described in Section 2). A representative WB with Coomassie blue staining (**left** panel) and the corresponding densitometric analysis (**right** panel) are reported. Data in the bar diagram are normalized to the mean value of control (G93A) myocytes. * *p*-value < 0.05, n.s., not significant, one-way ANOVA followed by Sidak’s multiple comparison test. (**B**,**C**) Differentiation index (**B**) and fusion index (**C**) in hSOD1(G93A) myocytes (4 DIV) cultured in the absence or the presence of MCM(WT). *n* = 5 *, *p*-value < 0.05, Mann–Whitney *U* test.

**Figure 4 cells-12-02751-f004:**
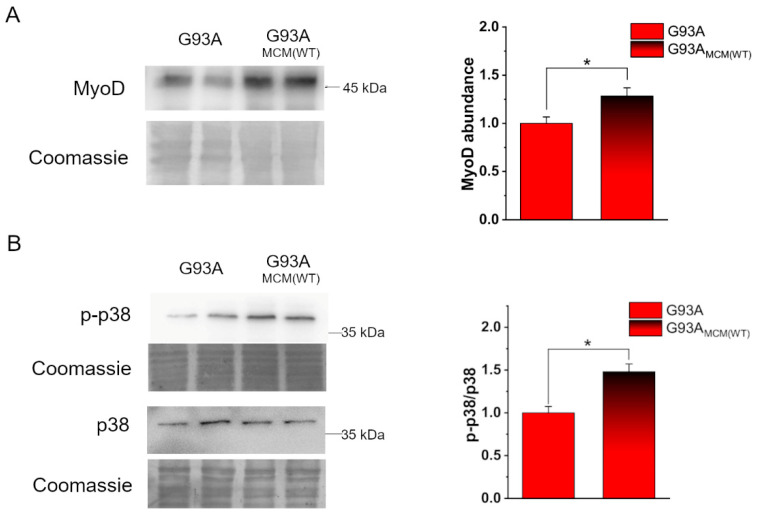
Myocyte-conditioned medium (MCM) from hSOD1(WT) myocytes recovers the expression of MyoD and the activation of p38 in hSOD1(G93A) cells. (**A**) Analysis of MyoD abundance in hSOD1(G93A) myocytes grown 2 DIV in the absence or the presence of hSOD1(WT) (MCM(WT) as described in Section 2). On the left, a representative WB and the corresponding Coomassie staining, while on the right, the densitometric analysis is reported. (**B**) WB analysis of p38 activation in the two different conditions (as in (**A**)). A representative WB of phosphorylated p38 (p-p38) and total p38, with the corresponding Coomassie blue staining are reported on the left. On the right, the corresponding densitometric analysis is reported. Data in the bar diagram are normalized to the mean value of control (G93A) myocytes. *n* = 6. * *p*-value < 0.05, Mann–Whitney *U* test.

**Figure 5 cells-12-02751-f005:**
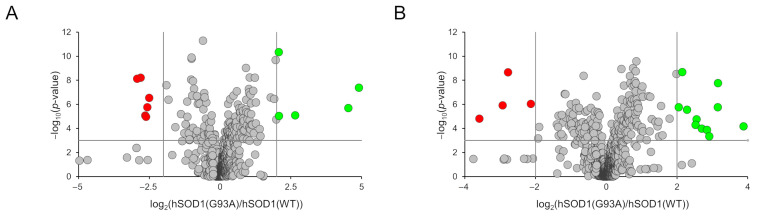
Volcano plots depicting the distribution of the 1297 and 1141 compounds detected in the positive (**A**) and negative (**B**) ionization polarities. The representing criteria for the *x* and *y* axes are the abundance of metabolites (expressed as the log_2_ (G93A vs. WT ratio)) and the statistical significance (expressed as the −log_10_ (*p*-value)), respectively. Cut-off parameters are shown as vertical and horizontal grey lines. Grey circles, not altered metabolites; green circles, over-represented metabolites in hSOD1(G93A); red circles, under-represented metabolites in hSOD1(G93A).

**Figure 6 cells-12-02751-f006:**
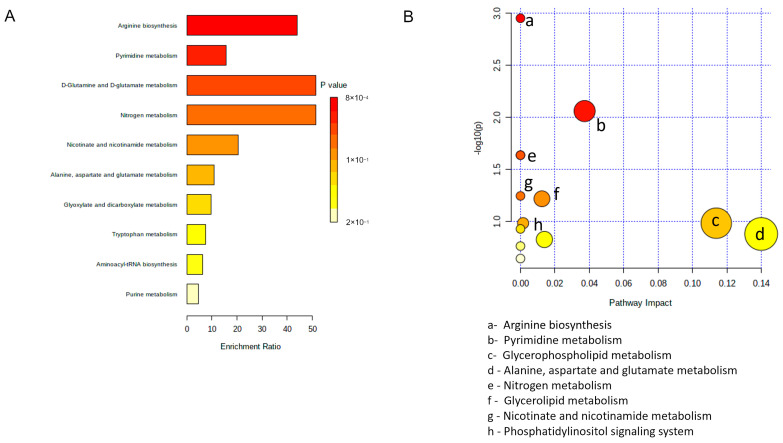
Pathway analysis of altered metabolites in hSOD1(G93A) MCM. (**A**) Metabolite set enrichment analysis using small molecule pathway database. (**B**) Metabolomic pathway analysis using the KEGG database.

**Table 1 cells-12-02751-t001:** List of altered metabolites in MCM of hSOD1(G93A) primary cultured myocytes. For each metabolite, the observed *m*/*z* value, chromatographic retention time, ionization polarity, proposed adduct, calculated molecular formula, molecular weight, possible identification, ratio value, and statistical significance are reported.

*m*/*z*	Retention Time (min)	Adduct	Formula	Polarity	Annotation	Ratio [hSOD1(G93A)/hSOD1(WT)]	*p*-Value
81.0452	9.31	[M+H]^+^	C_4_H_4_N_2_	positive	Pyrimidine	0.17	1.70 × 10^−6^
95.0607	7.62	-	-	positive	-	0.14	6.20 × 10^−9^
121.0648	7.09	[M+H]^+^	C_8_H_8_O	positive	4-Hydroxystyrene	0.13	7.44 × 10^−9^
124.0394	4.88	[M+H]^+^	C_6_H_5_NO_2_	positive	Nicotinic acid	4.20	4.61 × 10^−11^
127.0503	9.31	[M+H]^+^	C_5_H_6_N_2_O_2_	positive	Thymine	0.18	2.87 × 10^−7^
129.0659	4.74	[M+H]^+^	C_5_H_10_N_2_O_3_	positive	L-Glutamine	0.16	1.08 × 10^−5^
157.0970	7.69	[M-H_2_O+H]^+^	C_7_H_14_N_2_O_3_	positive	N2-Acetyl-L-ornithine	0.16	8.57 × 10^−6^
160.0756	7.57	[M+H]^+^	C_10_H_9_NO	positive	Indoleacetaldehyde	30.04	4.15 × 10^−8^
160.0756	11.64	[M+H]^+^	C_10_H_9_NO	positive	Indoleacetaldehyde	23.15	2.06 × 10^−6^
308.1925	7.59	[M+NH_4_]^+^	C_11_H_22_N_4_O_5_	positive	Ser Lys Gly	6.29	8.23 × 10^−6^
392.1812	5.33	[M-H_2_O+H]^+^	C_19_H_27_N_3_O_7_	positive	Glu Tyr Val/Tyr Asp Ile	4.24	9.10 × 10^−6^
81.0457	9.29	[M-H]^−^	C_4_H_6_N_2_	negative	4-Methylimidazole	0.23	9.17 × 10^−7^
161.0123	9.31	[M+Cl]^−^	C_5_H_6_N_2_O_2_	negative	Thymine	0.08	1.55 × 10^−5^
331.0784	4.99	-	-	negative	-	4.87	2.83 × 10^−6^
354.2032	4.75	[M-H_2_O-H]^−^	C_17_H_31_N_3_O_6_	negative	Glu Leu Ile	6.49	1.04 × 10^−4^
374.1719	4.90	[M-H_2_O-H]^−^	C_19_H_27_N_3_O_6_	negative	Phe Asp Leu/Pro Pro Tyr	8.89	1.70 × 10^−6^
374.1725	1.49	[M-H]^−^	C_19_H_25_N_3_O_5_	negative	Phe Asp Leu/Pro Pro Tyr	4.43	2.04 × 10^−9^
390.1800	4.76	[M-H_2_O-H]^−^	C_23_H_27_N_3_O_4_	negative	Phe Pro Phe	7.53	4.43 × 10^−4^
391.2260	1.45	[M+Cl]^−^	C_20_H_36_O_5_	negative	PGF1α	0.13	1.22 × 10^−6^
396.1000	6.00	[M+Cl]^−^	C_14_H_23_N_3_O_6_S	negative	Met Asp Pro	5.76	5.08 × 10^−5^
410.1486	4.89	[M+Cl]^−^	C_19_H_25_N_3_O_5_	negative	Pro Pro Tyr	14.70	6.57 × 10^−5^
419.2568	1.45	[M−H_2_O−H]^−^	C_21_H_43_O_7_P	negative	LPA(0:0/18:0)	0.15	2.19 × 10^−9^
426.1439	5.33	[M+Cl]^−^	C_19_H_25_N_3_O_6_	negative	Pro Phe Glu	5.87	1.76 × 10^−5^
551.1812	4.89	[M-H_2_O-H]^−^	C_24_H_34_N_4_O_10_S	negative	Met Glu Tyr Glu	4.14	1.75 × 10^−6^
787.8521	7.69	-	-	negative	-	7.21	1.29 × 10^−4^

## Data Availability

All data generated or analyzed during this study are available from the corresponding author upon reasonable request.

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
