# Peer review of "Secreted Metabolome of ALS-Related hSOD1(G93A) Primary Cultures of Myocytes and Implications for Myogenesis"

_cells, 2023, doi:10.3390/cells12232751_

Round 1

Reviewer 1 Report

Comments and Suggestions for Authors

Stella et al., studied differentiation of human SOD1-G93A (hSOD1-G93A) and control hSOD1-WT mouse myocytes in vitro, and like others before, found that it is impaired in the former relative to the latter. They followed up this observation by analyzing the secreted metabolome of these cells using mass spectrometry, both at the proliferation and differentiation states. They found metabolite alterations in the former and not the later in hSOD1-G93A cells and were able to rescue their differentiation by supplying conditioned medium from proliferating hSOD1-WT cells. The significance of this research relates to mutations in hSOD1 as one of the main genetic causes behind familial ALS, a fatal adult motor neuron disease, currently with few really effective treatments. Muscle denervation seems to precede motor neuron loss in ALS, so understanding this process, as caused by local impairment in the synaptic region of skeletal muscle fibers, may lead to future more effective therapies.

The manuscript is well-written, the data appears solid and rigorous, figures and tables are generally clear (although see below), and the discussion is measured, pointing out derived new insights and limitations.

The following issues remain to be addressed:

Line 47-50: Neuromuscular junctions (NMJs) are now considered tripartite synapses, where motor neurons, skeletal muscle fibers and perisynaptic, terminal Schwann cells (tSCs), play all important roles in its maintenance. Setting up the loss of NMJs as either caused by neuronal or muscle leaves out a role for tSCs, which in the hSOD1 ALS model studied here has supporting evidence from multiple labs. This is an issue raised in the Introduction by the authors, but that perhaps, also merits some discussion in the Discussion section of the paper.

Line 120: Conditioned media appears to have been collected in the presence of sera. Could this have been done in serum-free media instead? Could this have affected the failure to find differences between the conditioned media during the differentiation phase?

Line 197: By precipitating out proteins from conditioned media for metabolome analysis, authors may have removed the most interesting components, perhaps the ones truly responsible for the effects in their rescue experiments (Fig 3).

Line 259: Do you mean bright-field and not light-field images?

Figure 1: What are the differences between the representative images the authors want the reader to see? No myotubes in the bottom panel for the mutant? It is not clear. Also, what are the units in the X-axis for panel B?

Line 315: The rescue of hSOD1-G93A cell differentiation by the conditioned media from proliferating hSOD1-WT cells is a nice result. I wonder if the opposite experiment, trying to impair differentiation of hSOD1-WT cells with conditioned media from hSOD1-G93A cells was ever attempted by the authors? If not or if so, please expand in your reply.

Reviewer 2 Report

Comments and Suggestions for Authors

Aim of this work was to analyse the ALS skeletal muscle secreted metabolome. To this aim, the Authors used isolated primary cultured skeletal myocytes and performed a comparative characterization of hSOD1(G93A) and hSOD1(WT) myocyte cultures by LC-HRMS at the end of the proliferation step and after differentiation. The Authors reported that hSOD1(G93A)-derived culture medium has a different secretome compared to the hSOD1(WT); moreover, the addition of healthy myocyte conditioned medium seems to be able to correct the delayed in vitro myogenic program of ALS cells.

The work is very interesting because it shed light on the role of the skeletal muscle secretome in the regulation of myogenic program and into mechanisms of ALS pathogenesis and progression. The work is well written, with a precise description of the methods and an adequate analysis of the data.

Minor concern

1.       Minor typing errors are present in the text, I recommend a check of the manuscript;

2.       Figure 1 – Panel A. I suggest the Authors insert the scale bar also in the image “hSOD1(G93A)”;

3.       Figure 1 - Panel B. I suggest the Authors insert "Frequency distribution" on the y-axis instead of just "Frequency";

4.       Figure 1 – I suggest the Authors define “ΔA” in the figure legend

5.       I suggest the Authors improve (if possible) the representative WB stained with Coomassie blue.

Round 2

Reviewer 1 Report

Comments and Suggestions for Authors

Authors have responded to my concerns